# *Toxoplasma gondii* eIF-5A Modulates the Immune Response of Murine Macrophages In Vitro

**DOI:** 10.3390/vaccines12010101

**Published:** 2024-01-19

**Authors:** Xinchao Liu, Xiaoyu Li, Chunjing Li, Mingmin Lu, Lixin Xu, Ruofeng Yan, Xiaokai Song, Xiangrui Li

**Affiliations:** 1Anhui Province Key Laboratory of Animal Nutritional Regulation and Health, College of Animal Science, Anhui Science and Technology University, Fengyang 233100, China; liuxch@ahstu.edu.cn; 2MOE Joint International Research Laboratory of Animal Health and Food Safety, College of Veterinary Medicine, Nanjing Agricultural University, Nanjing 210095, China; 2016107054@naju.edu.cn (X.L.); 2017107054@naju.edu.cn (C.L.); mingmin.lu@njau.edu.cn (M.L.); xulixin@njau.edu.cn (L.X.); yanruofeng@njau.edu.cn (R.Y.); songxiaokai@njau.edu.cn (X.S.)

**Keywords:** *Toxoplasma gondii*, eukaryotic translation initiation factor 5A, immune response, macrophages, in vitro

## Abstract

*Toxoplasma gondii* (*T. gondii*) is an obligate intracellular protozoan that can elicit a robust immune response during infection. Macrophage cells have been shown to play an important role in the immune response against *T. gondii*. In our previous study, the eukaryotic translation initiation factor 5A (eIF-5A) gene of *T. gondii* was found to influence the invasion and replication of tachyzoites. In this study, the recombinant protein of *T. gondii* eIF-5A (r*Tg*eIF-5A) was incubated with murine macrophages, and the regulatory effect of *Tg*eIF-5A on macrophages was characterized. Immunofluorescence assay showed that *Tg*eIF-5A was able to bind to macrophages and partially be internalized. The Toll-like receptor 4 (TLR4) level and chemotaxis of macrophages stimulated with *Tg*eIF-5A were reduced. However, the phagocytosis and apoptosis of macrophages were amplified by *Tg*eIF-5A. Meanwhile, the cell viability experiment indicated that *Tg*eIF-5A can promote the viability of macrophages, and in the secretion assays, *Tg*eIF-5A can induce the secretion of interleukin-6 (IL-6), tumor necrosis factor-α (TNF-α) and nitric oxide (NO) from macrophages. These findings demonstrate that eIF-5A of *T. gondii* can modulate the immune response of murine macrophages in vitro, which may provide a reference for further research on developing *T. gondii* vaccines.

## 1. Introduction

*T. gondii* is an important intracellular protozoan parasite of humans and animals with a global distribution [1,2]. It can remarkably invade and replicate in nearly all nucleated cells [3]. In human hosts, toxoplasmosis is generally asymptomatic, but acute infection or reactivation of encysted parasites is a frequent cause of morbidity and mortality in immunocompromised individuals and the developing fetus [4]. In animals, *T. gondii* had a broad range of hosts, including wild and domestic animals. The infection of *T. gondii* has caused substantial losses due to abortion or the production of milk and meat in livestock, but there are still no effective human vaccines or new drugs to prevent or treat congenital and chronic infections [5,6]. Thus, toxoplasmosis is one of the most globally significant zoonotic diseases.

*T. gondii* can persist within cells for the whole life of the host, which requires a suitable ability to resist host cell defenses so that it can spread within its hosts without killing them. This balance between *T. gondii* and its host is important for the survival of both the parasite and the host [7]. By comprehending how the parasite and host establish this equilibrium, we can employ suitable methods to disrupt the balance and enhance the host’s anti-*T. gondii* response, thereby eliminating the infection.

Since *T. gondii* is an obligate intracellular parasite, the viability and apoptosis of host cells are extremely important for its survival [8]. Furthermore, after the establishment of infection, the host’s immune response is activated, which is critical to clear the parasites or strike the balance between *T. gondii* and the host [8,9]. Cell-mediated immunity plays an important role in host resistance to *T. gondii* infection. Macrophage is one of the effector cells mediating the host responses against *T. gondii* infection [10]. Phagocytosis, antigenic presentation and the production of proinflammatory cytokines are the main actions of macrophage resistance to *T. gondii*. Therefore, studying the interaction between *T. gondii* and macrophages is of great significance for understanding the balanced relationship between the parasite and the host.

The eukaryotic translation initiation factor 5A (eIF-5A) is an important translation factor found in all eukaryotic organisms. In addition to their regulatory roles in the initiation and elongation of proteins, studies have shown that eIF-5A also contributes to the proliferation, migration and invasive capacity of cancer cells [11,12,13]. In *T. gondii*, eIF-5A is an essential protein for the survival of tachyzoites [14]. Furthermore, eIF-5A indirectly affects the invasion of tachyzoite by regulating the expression of microneme proteins (MICs), rhoptry proteins (ROPs) and dense granule antigens (GRAs), and it also participates in the replication process of parasites through the ribosome pathway [14]. In summary, *Tg*eIF-5A plays an important role in *T. gondii* infection. Nevertheless, whether eIF-5A affects the immune response in host–parasite interplay remains unknown. In this study, macrophages were selected as the representative immune cells to investigate the effect of *Tg*eIF-5A on the immune response against *T. gondii*, thus offering insights for the development of *T. gondii* vaccines.

## 2. Materials and Methods

### 2.1. Cell Culture

The mouse macrophages (cell line: Ana-1, NCACC; CRL-GNM 2) were used in this study. The cells were maintained in Dulbecco’s modified Eagle’s medium (DMEM, Invitrogen Life Technologies, Carlsbad, CA, USA) supplemented with 10% dialyzed fetal bovine serum (FBS, Invitrogen Life Technologies, Carlsbad, CA, USA) and 1% penicillin–streptomycin (Invitrogen Life Technologies, Carlsbad, CA, USA) in a CO_2_ incubator (Thermo Fisher Scientific, Waltham, MA, USA) at 37 °C.

### 2.2. Recombinant Protein and Polyclonal Antibody of TgeIF-5A

The recombinant protein of *Tg*eIF-5A fused with polyhis-tag (r*Tg*eIF-5A), the protein encoded by the polyhis-tag of the empty pET-32a (+) vector (pET-32a vector protein), and a polyclonal antibody against eIF-5A were obtained in the previous study and stored in our laboratory [14]. Briefly, the open reading frame of *Tg*eIF-5A was amplified by PCR and cloned into prokaryotic expression vector pET-32a. The recombinant vector was transformed into *Escherichia coli* (*E. coli*) BL21 and induced with isopropyl β-d–thiogalactopyranoside (IPTG, Sigma-Aldrich, St. Louis, USA) to obtain the recombinant protein. A Ni^2+^-nitrilotriacetic acid (Ni-NTA) column (GE Healthcare, Madison, WI, USA) was used to purify the r*Tg*eIF-5A protein. The purified r*Tg*eIF-5A protein was emulsified with Freund’s complete or incomplete adjuvant (Beyotime, Shanghai, China) to immunize rats to prepare the polyclonal antibody against *Tg*eIF-5A.

### 2.3. Immunofluorescence and Confocal Microscopy

A total of 1 × 10^5^ Ana-1 cells were seeded in a 24-well plate and stimulated with 20 μg/mL r*Tg*eIF-5A for 1 h in a binding assay and 24 h in an internalization assay, followed by collection in a 1.5 mL centrifuge tube. At each time point, the cells were rinsed with phosphate-buffered saline (PBS) 3 times. The cells were transferred to poly-l-lysine-coated glass slides and then fixed with 4% paraformaldehyde in PBS for 30 min at 37 °C. The glass slides were kept in a blocking solution (5% bovine serum albumin, BSA) for 1 h and incubated with the rat anti-r*Tg*eIF-5A antibody diluted in 5% BSA at 4 °C for 12 h. Then, goat anti-rat IgG antibody labeled with Cyanine 3 (Cy3, Beyotime, Shanghai, China) was used as the secondary antibody to incubate with the cells for 1 h at 37 °C. The nuclei were stained with 4′,6-diamidino-2-phenylindole (DAPI, Beyotime, Shanghai, China) for 5 min. The control and blank groups (stimulated with pET-32a vector protein or PBS) received the same treatment as the experimental group simultaneously. Finally, the cells were fixed using Fluoromount–G (Beyotime, Shanghai, China) and visualized at 555/570 nm (Ex/EM) for Cy3 and 360/460 nm (Ex/EM) for DAPI using a confocal microscope (Zeiss, Jena, Germany, 630× magnification).

### 2.4. Cell Viability Assay

For the measurement of cell viability, 5 × 10^5^ Ana-1 macrophages were cocultured with r*Tg*eIF-5A proteins at concentrations of 5, 10, 20, 40 or 80 µg/mL, pET-32a vector proteins or PBS in a 96-well plate for 24 h, respectively. A total of 10 μL of Cell Counting Kit 8 reagent (CCK-8, Beyotime, Shanghai, China) was added to each well and cocultured with the stimulated or unstimulated cells for 1 h. The cell viability of Ana-1 macrophage cells was measured using a microplate spectrophotometer (BioRad, Hercules, CA, USA) at an optical density of 450 nm.

### 2.5. Migration Assays

For migration assays, 5 × 10^6^ Ana-1 macrophages were stimulated with r*Tg*eIF-5A proteins at concentrations of 0, 5, 10, 20, 40 or 80 µg/mL or pET-32a vector protein for 24 h in a 24-well plate (Corning Costar, Cambridge, MA, USA), respectively. The cocultured cells were inserted into Millicell Hanging Cell Culture Inserts (Merck Millipore, Billerica, MA, USA) and allowed to migrate toward monocyte chemoattractant protein (MCP-1 or CCL2, Medchemexpress, Shanghai, China) or no chemokines for 24 h. The migrated cells were counted under a light microscope, and the migration rate was analyzed.

### 2.6. Flow Cytometric Assessment of Apoptosis, Phagocytosis and Immunophenotyping

Ana-1 cell suspensions were seeded in 12-well plates (Corning Costar, Cambridge, MA, USA) and incubated with r*Tg*eIF-5A proteins at concentrations of 0, 5, 10, 20, 40 or 80 µg/mL, pET-32a vector protein or PBS for 24 h in a CO_2_ incubator at 37 °C, respectively. The washed cells of each well were divided into three fluorescence-activated cell sorting (FACS) tubes and stained with the following reagents in different experiments: fluorescein isothiocyanate (FITC)—labeled Annexin V and propidium iodide (PI) (Miltenyi Biotec, Bergisch Gladbach, Germany) for apoptosis, FITC—labeled dextran for phagocytosis (Sigma-Aldrich, St. Louis, MO, USA) and phycoerythrin (PE)—labeled anti-CD284 antibody (clone: SA15-21, Toll-like receptors 4, TLR4, Biolegend, San Diego, CA, USA) for the detection of the TLR4 level of macrophages. The apoptosis of Ana-1 cells was distinguished into early apoptosis (annexin V positive and PI negative) and late apoptosis (annexin V and PI positive), and the cells boiled in water for 10 s were used as a positive control. The cells were acquired on a FACScan™ device (BD Biosciences, San Jose, CA, USA). The data were analyzed using FlowJo software (Version 9.0, Tree Star, OR, USA) and displayed as frequencies or median fluorescence intensity (MFI).

### 2.7. Secretion Assays

Ana-1 macrophages were stimulated with r*Tg*eIF-5A proteins at concentrations of 0, 5, 10, 20, 40 or 80 µg/mL, pET-32a vector protein or PBS for 24 h in a 24-well plate, respectively. The culture supernatants were collected. The levels of tumor necrosis factor-α (TNF-α), interleukin-1β (IL-1β), IL-6, IL-10 and IL-12 were measured in the culture supernatants of Ana-1 macrophages stimulated with r*Tg*eIF-5A proteins, pET-32a vector protein or left unstimulated using a cytometric bead array kit (CBA kit, BD Biosciences, San Jose, CA, USA). The levels of nitric oxide (NO) in the supernatants were measured by a Nitric Oxide Assay Kit (Beyotime, Shanghai, China).

### 2.8. Statistics

The Kolmogorov–Smirnov test was used to assess the normality of data distribution. Statistical analysis was performed using GraphPad Prism (Version 9.0, GraphPad Software, La Jolla, CA, USA). One-way ANOVA followed by Tukey’s multiple comparisons was used to analyze the differences between groups. Data are representative of three independent experiments and shown as the means ± the standard deviation (SD). Differences were considered significant when *p* < 0.05 (*, 0.01< *p* < 0.05, **, 0.001< *p* < 0.01, ***, 0.0001< *p* < 0.001 and ****, *p* < 0.0001).

## 3. Results

### 3.1. TgeIF-5A Binds to and Internalizes into Macrophages

r*Tg*eIF-5A proteins were added to the medium to their interaction with macrophages was visualized. We observed that r*Tg*eIF-5A bound to the surface of Ana-1 cells and exhibited red fluorescence resulting from Cy3, and after 24 h of culture, some r*Tg*eIF-5A proteins were internalized into Ana-1 cells. On the other hand, on the control cells without r*Tg*eIF-5A proteins, only blue fluorescence (DAPI) exhibited from the cell nucleus was observed (Figure 1).

### 3.2. TgeIF-5A Reduces the TLR4 Level of Macrophages

The binding and internalization to Ana-1 cells suggest that *Tg*eIF-5A might contribute to the modulation of cellular functions. To address this, we determined the impact of eIF-5A on the innate immune defense of macrophages. TLRs are a family of membrane-bound receptors that can recognize “non-self” molecules during infection [15]. We therefore assessed the TLR4 levels of macrophages cocultured with r*Tg*eIF-5A. We observed that the levels of TLR4 in r*Tg*eIF-5A-stimulated Ana-1 cells were significantly reduced compared to the pET-32a vector protein control group and unstimulated macrophages (Figure 2). Notably, we also observed that the reduction in TLR4 level was related to the concentration of r*Tg*eIF-5A. The higher the concentration of r*Tg*eIF-5A stimulated, the more TLR4 levels were reduced.

### 3.3. TgeIF-5A Induces Higher Levels of TNF-α, IL-6 and NO Secretion in Macrophages

Cytokines are key factors in the immune system. They are secreted from various types of immune cells to manipulate immune cell physiology and counteract pathogens [16]. To investigate whether eIF-5A affects the secretion of Ana-1 cells, we evaluated the secretion of TNF-α, IL-1β, IL-6, IL-10 and IL-12 in Ana-1 cells stimulated with r*Tg*eIF-5A or pET-32a vector protein, as well as in unstimulated macrophages, using CBA cell signaling flex sets. We found that the secretions of TNF-α and IL-6 were amplified significantly in Ana-1 cells cocultured with r*Tg*eIF-5A (Figure 3). In addition, Ana-1 cells produced higher levels of TNF-α and IL-6 after being stimulated with high concentrations of eIF-5A. However, there were no differences in the secretion levels of IL-1β, IL-10 and IL-12 in macrophages stimulated with r*Tg*eIF-5A, pET-32a vector protein and those left unstimulated.

The production of NO is another mechanism to limit parasite growth [17]. Therefore, the secretion of NO in Ana-1 cells was investigated. We observed that the levels of NO in Ana-1 cells stimulated with r*Tg*eIF-5A were significantly higher than those stimulated with pET-32a vector protein or left unstimulated (Figure 3).

### 3.4. TgeIF-5A Inhibits the Migration of Macrophages

Migration is the primary condition for the immune activity of macrophages to defend against infection. To determine whether eIF-5A contributes to the migration of Ana-1 cells, a chemotaxis assay was determined in eIF-5A or pET-32a vector protein-stimulated cells with MCP-1 or without MCP-1. Notably, in the Ana-1 cells stimulated with pET-32a vector protein, we observed a significant increase in the migration rate of Ana-1 cells toward MCP-1 (Control group) compared to the cells without MCP-1 (Blank group, Figure 4A). In addition, the migration was reduced significantly in Ana-1 cells stimulated with eIF-5A compared to pET-32a vector protein towards MCP-1.

### 3.5. TgeIF-5A Induces the Phagocytosis of Macrophages

The phagocytic function of macrophages plays a pivotal role in combating invading pathogens. To investigate a possible effect of eIF-5A on the phagocytosis of macrophages, we incubated FITC-dextran with r*Tg*eIF-5A-treated Ana-1 cells, which resulted in the amplification of phagocytosis relative to the pET-32a vector protein control group and unstimulated group. As shown in Figure 4B, we observed that 10 μg/mL of eIF-5A protein reduced the most robust phagocytic ability of macrophages.

### 3.6. TgeIF-5A Induces the Apoptosis of Macrophages

Inhibiting host cell apoptosis is a strategy employed by intracellular pathogens to ensure their own survival [8]. To determine whether *Tg*eIF-5A affects the apoptosis of macrophages, we cocultured r*Tg*eIF-5A with Ana-1 cells and stained the cells with FITC-labeled annexin V and PI. Flow cytometry analysis revealed that the percentage of apoptotic macrophage cells in the unstimulated group (median, 2.81% early apoptosis and 0.78% late apoptosis) was similar to that of macrophages stimulated with pET-32a vector protein (median, 3.50% early apoptosis and 0.91% late apoptosis). r*Tg*eIF-5A-stimulated macrophages showed significantly increased levels of both early (median, 7.35%) and late stages of apoptosis (median, 1.69%) than unstimulated ones (Figure 5).

### 3.7. TgeIF-5A Induces the Cell Viability of Macrophages

To analyze whether eIF-5A affects the viability of macrophages, a CCK8 assay was performed in Ana-1 cells stimulated with r*Tg*eIF-5A, pET-32a vector protein or left unstimulated for 24 h, followed by incubation with CCK8 for 1 h. We observed that the viability of Ana-1 cells cocultured with high concentrations of r*Tg*eIF-5A was significantly amplified relative to the unstimulated macrophages (Figure 6).

## 4. Discussion

In response to *T. gondii* infection, the immune system of the host builds a protective immune response. Immune cells, including macrophages, dendritic cells and natural killer (NK) cells, interact in a coordinated way to form the innate immune response against *T. gondii* [10]. Macrophages are an important component of the innate immune response. They are rapidly recruited to the sites of infections and trigger a variety of mechanisms to eliminate and control *T. gondii* such as directly phagocytizing the parasite and releasing cytokines to develop adaptive immune response [8,18].

For successful transmission, *T. gondii* has to balance the host immune response to establish infection [19]. For instance, the excretory/secretory antigens (ESAs) of *T. gondii* can inhibit TLR-induced nuclear factor kappa B (NF-κB) activation, thereby suppressing the secretion of proinflammatory cytokines [20]. *T. gondii* ROP16 inhibits the secretion of IL-12 by activating the signal transducer and activator of transcription 3 (STAT3) and STAT6 [21]. *T. gondii* GRA15 plays an important role in activating the NF-κB pathway by interacting with TNF receptor-associated factors (TRAFs) [22]. *T. gondii* ROP18 targets host cell proapoptotic protein purinergic receptor 1 (P2X1), which leads to the inhibition of host cell apoptosis [23]. These results indicate that the parasite proteins are involved in the modulation of host cellular events associated with the host’s immune responses. Among *T. gondii* proteins, eIF-5A has been shown recently to be involved in the regulation of *T. gondii* invasion and replication [14]. Here, we report that eIF-5A of *T. gondii* is a binding protein of macrophages, and after co-incubation with macrophages, a portion of eIF-5A is internalized into macrophages. Studies have shown that when parasite proteins are internalized into host cells, they can affect host cell signaling pathways and immune responses [24,25]. Therefore, whether eIF-5A participates in the regulation of macrophage’s immune responses is also demonstrated in this study.

The innate immunity against *T. gondii* is mediated by the interaction of TLRs with the ligands expressed on the *T. gondii* surface, and previous studies have shown that *T. gondii* MIC1 and MIC4 interact with TLR2 and TLR4 N-glycans to trigger IL-12 responses [26]. Our findings show that eIF-5A reduces the production of TLR4 in macrophages, which suggests that eIF-5A can reduce the protective immunity against *T. gondii* by reducing the TLR4 production in macrophages. The release of IL-12 from macrophages stimulates T cells and NK cells to produce interferon-gamma (IFN-γ), which is a key mediator of resistance to either the acute or chronic infection of *T. gondii* infection [27,28]. We observed that there were no differences in IL-12 expression in macrophages exposed to eIF-5A and in those left unexposed, suggesting that although eIF-5A could reduce the production of TLR4 in macrophages, it cannot reduce the secretion of IL-12. Therefore, we speculate that eIF-5A may also induce the secretion of IL-12 through other pathways, which requires further investigation.

Other cytokines and inflammatory mediators also play important roles in *T. gondii* infection [10]. TNF-α, an inflammatory mediator produced by macrophages and other cells, has been found to be responsible for the production of acute inflammatory response [29]. In *T. gondii* infection, TNF-α synergizes with IFN-γ to play an important role in the protective immunity against toxoplasmosis [30,31]. Our findings indicate that eIF-5A contributes to TNF-α production in macrophages. Additionally, eIF-5A also induces high levels of IL-6 production that functions synergistically with TNF-α. However, the production of IL-1β, another cytokine that synergizes with TNF-α, is not affected by eIF-5A. These findings suggest a role of eIF-5A in modulating acute *T. gondii* infection. The production of NO is another antimicrobial activity of macrophages, which is implicated in the control of chronic *T. gondii* infection [32]. A previous study indicated that NO production was inhibited by *T. gondii* infection in different mouse macrophage cell lines [32]. We observed that eIF-5A induced the production of NO in Ana-1 macrophages, suggesting a role of eIF-5A in modulating chronic toxoplasmosis. As a result, the secretion assays showed that eIF-5A may have regulatory effects on both acute and chronic *T. gondii* infections.

When infection occurs in tissues, a variety of chemoattractants are produced from the pathogens and cells, leading to an increase in the migration of immune cells into the infection sites [33]. The migration of immune cells is one of the main components of inflammatory response [34]. Here, we report that the migration capability of macrophages is inhibited after stimulation with eIF-5A, suggesting that eIF-5A can inhibit the elimination of *T. gondii* by inhibiting the migration of macrophages. After macrophages recognize the pathogen as “foreign”, they will attempt to eliminate the pathogen [35]. Therefore, we further analyzed the effect of eIF-5A on the phagocytic ability of macrophages. We found that the phagocytosis of macrophages stimulated by eIF-5A was not inhibited but was enhanced. In particular, 10 μg/mL eIF-5A protein was most effective in inducing the phagocytic ability of macrophages, suggesting that this concentration was the optimal concentration for eIF-5A protein to induce macrophage phagocytosis. In contrast, the induction effect of eIF-5A protein on phagocytosis was not as robust at other concentrations. These results suggest that the regulatory effect of eIF-5A on the immune function of macrophages is complex, and further research is needed to clarify the specific mechanism.

Apoptosis is a form of programmed cell death that has an important role in the resistance to intracellular parasite infections [36]. To ensure that the parasite can complete its complex life cycle and transmit in different hosts, *T. gondii* has evolved a wide array of mechanisms to arrest apoptosis and maintain the viability of host cells [37]. On the other hand, the molecular mechanisms of *T. gondii* to arrest the host cell apoptosis are not clear yet. In this study, we observed that early apoptosis and late-stage apoptosis were induced in macrophages stimulated with eIF-5A compared to unstimulated controls, suggesting that eIF-5A plays a role in modulating host cell apoptosis. However, this modulation is detrimental to *T. gondii* replication and survival in host cells. When *T. gondii* infection results in the damage of host cells, the host cells need to be replaced in a timely manner. Thus, the cell viability of host cells is also fundamental in the response to *T. gondii* infection. We found that the viability of macrophages was induced by the stimulation of eIF-5A, suggesting that eIF-5A can modulate host cell viability to benefit the anti-*T. gondii* response. It is noteworthy that low concentrations of eIF-5A have no effects on the viability of macrophages but significantly promote late apoptosis. When high concentrations of eIF-5A increase the viability of macrophages, the number of late apoptotic cells decreases. This suggests that eIF-5A plays a certain regulatory role in cell viability and apoptosis, and the specific mechanism still needs further study.

## 5. Conclusions

*T. gondii* eIF-5A is a protein that can bind to and be internalized into macrophages, which can reduce the immune response against *T. gondii* by reducing the TLR4 production and migration of macrophages. Furthermore, eIF-5A can enhance the anti-*T. gondii* response of macrophages by promoting phagocytosis, apoptosis and viability as well as the release of TNF-α, IL-6 and NO. These results suggest that *T. gondii* eIF-5A is involved in modulating the immune response of murine macrophages in vitro and plays a more important regulatory role in enhancing the immune response of macrophages against *T. gondii*. Therefore, recombinant protein vaccines or nanovaccines based on eIF-5A might be helpful in preventing and controlling toxoplasmosis, but further research is needed to verify this conclusion.

## Figures and Tables

**Figure 1 vaccines-12-00101-f001:**
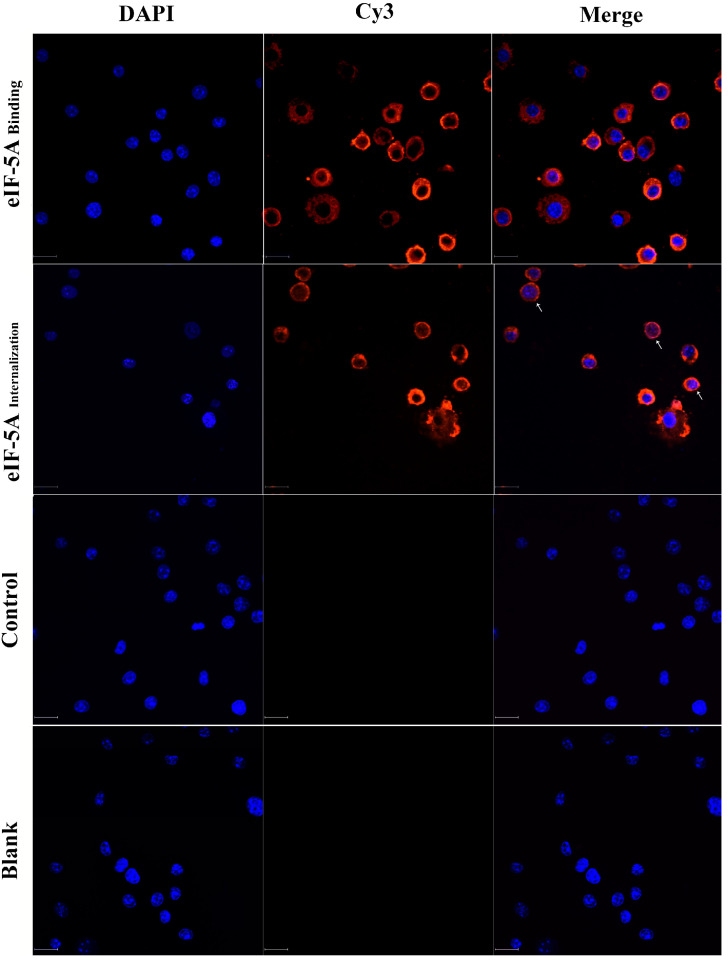
The binding and internalization of eIF-5A to macrophages. Macrophages were cocultured with 20 μg/mL eIF-5A protein, pET-32a vector protein (Control) or PBS (Blank). Cells were stained with Cy3-conjugated secondary antibody to visualize eIF-5A protein (red) and DNA was stained with DAPI (blue). Merge, overlapping the blue channels with red channels. Bars, 15 µm.

**Figure 2 vaccines-12-00101-f002:**
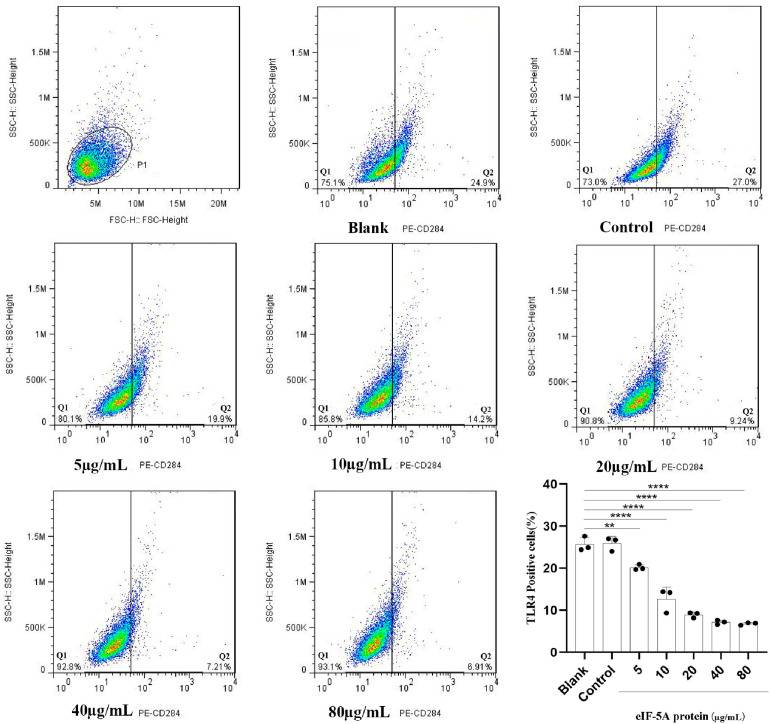
eIF-5A reduces the TLR4 production of murine macrophages. The TLR4 levels were measured in macrophages stimulated with eIF-5A protein, pET-32a vector protein (Control) or left unstimulated (Blank). The differences were relative to the values obtained with the unstimulated macrophages, and there was no difference between the cells stimulated with pET-32a vector protein and left unstimulated. The data were indicative of three individual experiments. No symbol indicated that the difference was not significant, *p* > 0.05; **, 0.001 < *p* < 0.01; ****, *p* < 0.0001.

**Figure 3 vaccines-12-00101-f003:**
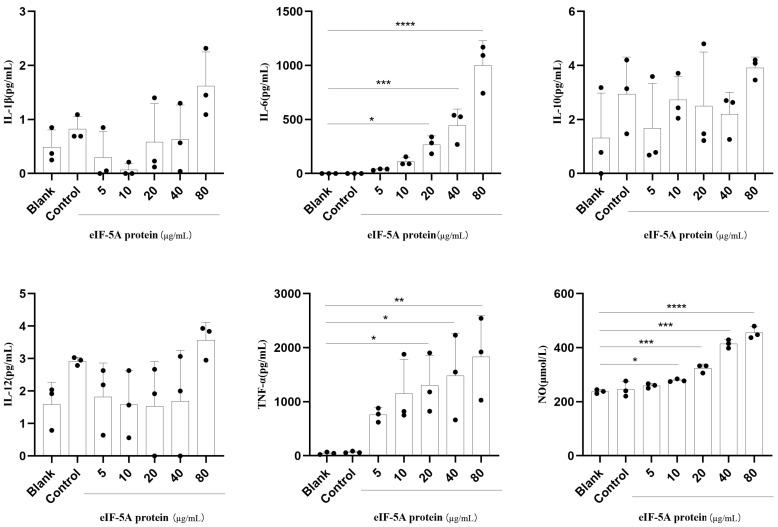
eIF-5A amplify the secretion of murine macrophages. The levels of IL-6, IL-10, TNF-α, IL-12 and IL-1β were assessed by CBA and the levels of NO were assessed by Total Nitric Oxide Kit in the supernatant macrophage culture stimulated with eIF-5A protein, pET-32a vector protein (Control) or left unstimulated (Blank) for 24 h. The differences were relative to the values obtained with the nonstimulated macrophages and there was no difference between the cells stimulated with pET-32a vector protein and left unstimulated. The data were indicative of three individual experiments. No symbol indicated that the difference was not significant, *p* > 0.05; *, 0.01 < *p* < 0.05; **, 0.001 < *p* < 0.01; ***, *p* < 0.001; ****, *p* < 0.0001.

**Figure 4 vaccines-12-00101-f004:**
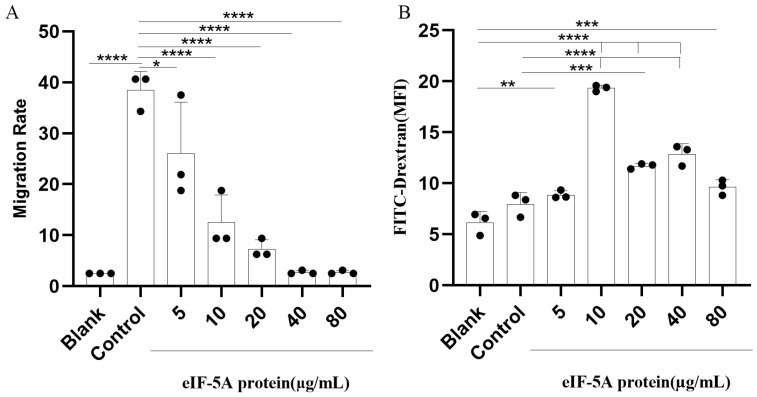
eIF-5A reduce the migration but induce the phagocytosis of macrophages. (**A**) Migration rates of macrophages stimulated with eIF-5A protein or pET-32a vector protein were assessed in a transwell plate toward MCP-1 or no MCP-1. The cells stimulated with pET-32a vector protein toward MCP-1 and no MCP-1 were set as control and blank group, respectively. The differences were relative to the values obtained with the macrophages stimulated with pET-32a vector protein towards MCP-1 (Control). (**B**) MFI of FITC-dextranin phagocytosed in macrophages stimulated with eIF-5A protein, pET-32a vector protein (Control) or left unstimulated (Blank). The differences were relative to the values obtained in both blank and control group. There was no difference between the cells stimulated with pET-32a vector protein and left unstimulated. The data were indicative of three individual experiments. No symbol indicated that the difference was not significant, *p* > 0.05; *, 0.01 < *p* < 0.05; **, 0.001 < *p* < 0.01; ***, 0.0001 < *p* < 0.001; ****, *p* < 0.0001.

**Figure 5 vaccines-12-00101-f005:**
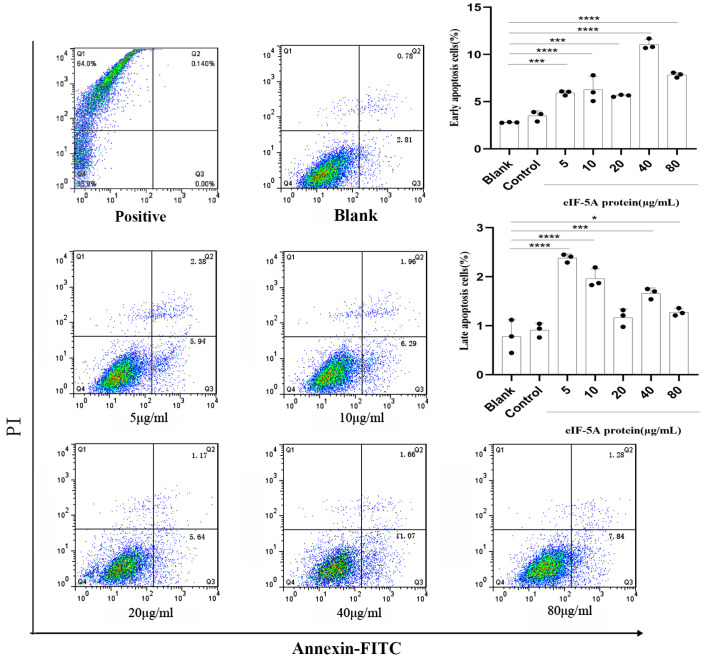
eIF-5A induced the apoptosis of murine macrophages. The apoptosis rates of macrophages stimulated with eIF-5A protein, pET-32a vector protein (Control) or left unstimulated (Blank) were assessed by flow cytometry using the Annexin V-FITC kit. The cells boiled in water for 10 s were used as a positive control. The differences were relative to the values obtained with the unstimulated macrophages and there was no difference between the cells stimulated with pET-32a vector protein and left unstimulated. The data were indicative of three individual experiments. No symbol indicated that the difference was not significant, *p* > 0.05; *, 0.01 < *p* < 0.05; ***, 0.0001 < *p* < 0.001; ****, *p* < 0.0001.

**Figure 6 vaccines-12-00101-f006:**
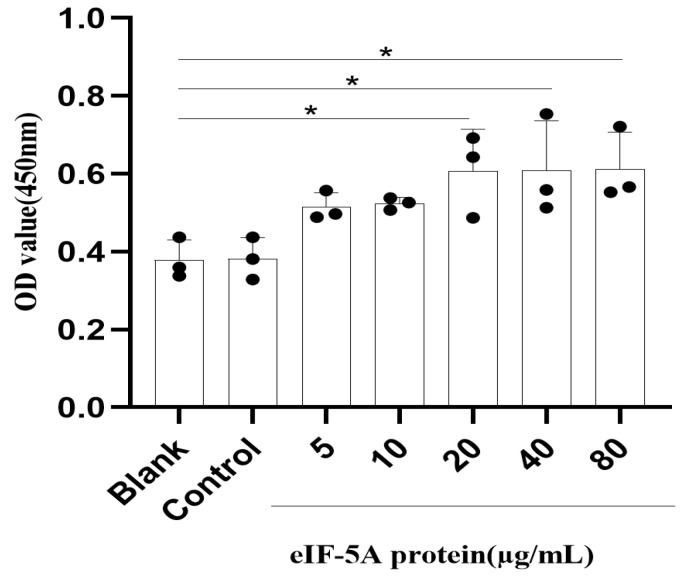
eIF-5A induced the cell viability of murine macrophages. The proliferation of macrophages stimulated with eIF-5A protein, pET-32a vector protein (Control) or left unstimulated (Blank) was assessed by CCK-8. The differences were relative to the values obtained with the unstimulated macrophages and there was no difference between the cells stimulated with pET-32a vector protein and left unstimulated. The data were indicative of three individual experiments. No symbol indicated that the difference was not significant, *p* > 0.05; *, 0.01 < *p* < 0.05.

## Data Availability

Data are contained within the article.

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
