# Peer review of "Toxoplasma gondii eIF-5A Modulates the Immune Response of Murine Macrophages In Vitro"

_vaccines, 2024, doi:10.3390/vaccines12010101_

Round 1

Reviewer 1 Report

Comments and Suggestions for Authors

Reviewer TR

Review Report for the article

Title:

Toxoplasma gondii eIF-5A Modulates the Immune Response of Murine Macrophages in vitro

Authors:

Xinchao Liu, Xiaoyu Li, Chunjing Li,  Mingmin Lu,  Lixin X,  Ruofeng Yan,  Xiaokai Song,  Xiang Rui Li

Report:

In this study, by using purified recombinant protein, the authors showed that the eukaryotic translation initiation factor 5A (eIF-5A) could bind to macrophages in cultured cells. The binding of this protein on macrophages caused a cascade of activities in the experimental system conducted by the study authors.  From these studies, the authors conclude that T. gondii eIF-5A is a macrophage-binding protein that can reduce the immune response against T. gondii by reducing the TLR4 production and migration of macrophages. Furthermore, eIF-5A can enhance the anti-T. gondii response of macrophages by promoting phagocytosis, apoptosis, proliferation, and the release of TNF-α, IL-6, and NO. These results suggest that T. gondii eIF-5A is involved in modulating the immune response of murine macrophages in vitro and might have important implications for toxoplasmosis prevention.

TgeIF-5A Binding Assay:

The authors analyzed the binding of TgeIF-5A protein to Ana-1 cells and for the binding assay, Ana-1 cells were seeded in a 24-well plate and stimulated with 20 μg/mL of rTgeIF-5A for 1 h.

For all other Assays:

The rTgeIF-5A protein was incubated with Ana-I cells for 24 hours.

Question/concern: The stability and the localization of the purified elF-5A in cell culture should be validated. There will be proteolytic degradation of the recombinant proteins in the culture system for a 24-hour incubation period. Authors should show that the protein is not degraded (by western blot) and that the localization of the recombinant protein in the sub-cellular compartments/organelles by immunofluorescence assays (IFA). Also, the authors did not show any clear experimental evidence showing that the mere binding and localization of eIF-5A on the plasma membrane of the cells is sufficient to cause all the activities presented in this report. The authors should design a clear experiment showing that protein is transported into the cell and binds to potential targets for all the activities presented in this study. The experimental results presented in this study are not sufficient for the conclusion of this study.

Throughout this study, the authors presented pET32-A vector protein- What is the vector protein? There should not be any vector protein purified through the Ni-NTA column.  This should be explained properly. There should not be any variation between blank and control. The authors should clearly define what is blank and how the blank is different from the control (I guess there should not be any variation between these arms of any experiment). Why there is so much variation between these two in several experiments? I am just curious to know the protein composition in the pET32-A vector protein mix. I assume the blank is just some buffer.

Also, for the binding assay, the authors used 20 20 μg/mL of rTgeIF-5A, and all other assays the concentration is 5-80 ng/ml. This is a concern because, at higher concentrations, the purified protein may non-specifically bind to the bind on to phospholipids of the membrane. This should be discussed.

Minor correction:

Line 221- As shown in Figure 4B, we observed that 10 μg/mL of eIF-5A protein had the most significant induction effect on the phagocytic ability of macrophages.

I think this should be 10 ng/ml.

Comments on the Quality of English Language

Some editing is required throughout the document. 

Author Response

Dear Reviewer:

Thank you for the constructive review and critical comments. We have uploaded the revised manuscript with highlighted changes. We believe that the manuscript is now much improved with changes made in response to the reviewers’ comments, and hopefully, we have already addressed all the comments and suggestions. Meanwhile, we have sought help from a native speaker for language editing. Thanks again for your consideration of our revised manuscript for publication in Vaccines.

Responses to comments from reviewers

Report:

In this study, by using purified recombinant protein, the authors showed that the eukaryotic translation initiation factor 5A (eIF-5A) could bind to macrophages in cultured cells. The binding of this protein on macrophages caused a cascade of activities in the experimental system conducted by the study authors.  From these studies, the authors conclude that T. gondii eIF-5A is a macrophage-binding protein that can reduce the immune response against T. gondii by reducing the TLR4 production and migration of macrophages. Furthermore, eIF-5A can enhance the anti-T. gondii response of macrophages by promoting phagocytosis, apoptosis, proliferation, and the release of TNF-α, IL-6, and NO. These results suggest that T. gondii eIF-5A is involved in modulating the immune response of murine macrophages in vitro and might have important implications for toxoplasmosis prevention.

TgeIF-5A Binding Assay:

The authors analyzed the binding of TgeIF-5A protein to Ana-1 cells and for the binding assay, Ana-1 cells were seeded in a 24-well plate and stimulated with 20 μg/mL of rTgeIF-5A for 1 h.

For all other Assays:

The rTgeIF-5A protein was incubated with Ana-I cells for 24 hours.

Question/concern: The stability and the localization of the purified elF-5A in cell culture should be validated. There will be proteolytic degradation of the recombinant proteins in the culture system for a 24-hour incubation period. Authors should show that the protein is not degraded (by western blot) and that the localization of the recombinant protein in the sub-cellular compartments/organelles by immunofluorescence assays (IFA). Also, the authors did not show any clear experimental evidence showing that the mere binding and localization of eIF-5A on the plasma membrane of the cells is sufficient to cause all the activities presented in this report. The authors should design a clear experiment showing that protein is transported into the cell and binds to potential targets for all the activities presented in this study. The experimental results presented in this study are not sufficient for the conclusion of this study.

Thank you for your suggestion. Based on your comments, immunofluorescence assays (IFA) was conducted to verify whether eIF-5A protein can be internalized into macrophages. The results show that some eIF-5A were internalized into macrophages (Figure 1, Lines 159-160). This part is also discussed in the revised article (Lines 296-299). According to another references, it can be inferred that eIF-5A can cause the activities presented in this report.

Ma'Ayeh SY, Liu J, Peirasmaki D,et al.Characterization of the Giardia intestinalis secretome during interaction with human intestinal epithelial cells:The impact on host cells[J].Plos Neglected Tropical Diseases, 2017, 11(12):e0006120.

Wang Y, Wu L, Liu X, Wang S, Ehsan M, Yan RF, Song XK, Xu LX, Li XR: Characterization of a secreted cystatin of the parasitic nematode Haemonchus contortus and its immune-modulatory effect on goat monocytes. Parasites & Vectors 2017, 10(1):425.

Throughout this study, the authors presented pET32-A vector protein- What is the vector protein? There should not be any vector protein purified through the Ni-NTA column. This should be explained properly. There should not be any variation between blank and control. The authors should clearly define what is blank and how the blank is different from the control (I guess there should not be any variation between these arms of any experiment). Why there is so much variation between these two in several experiments? I am just curious to know the protein composition in the pET32-A vector protein mix. I assume the blank is just some buffer.

Sorry for the confusion, and we have explained what is pET32-A vector protein (Lines 77-78) . 

Also, for the binding assay, the authors used 20 20 μg/mL of rTgeIF-5A, and all other assays the concentration is 5-80 ng/ml. This is a concern because, at higher concentrations, the purified protein may non-specifically bind to the bind on to phospholipids of the membrane. This should be discussed.

Sorry for the writing errors, and in all the assays except binding assay the concentration of rTgeIF-5A were 5-80 μg/mL.

Minor correction:

Line 221- As shown in Figure 4B, we observed that 10 μg/mL of eIF-5A protein had the most significant induction effect on the phagocytic ability of macrophages.

I think this should be 10 ng/ml.

Revised accordingly (Figure 2, 3, 4, 5, 6) .

Reviewer 2 Report

Comments and Suggestions for Authors

Dear Authors

the paper is of high scientific level and interest and it's well written. The only recommendation is for a minimal revision of the English form, especially for the use of adverbs

Comments on the Quality of English Language

as described above

Author Response

Dear reviewer:

Thank you for the constructive review and critical comments. We have uploaded the revised manuscript with highlighted changes. We believe that the manuscript is now much improved with changes made in response to the reviewers’ comments, and hopefully, we have already addressed all the comments and suggestions. Meanwhile, we have sought help from a native speaker for language editing. Thanks again for your consideration of our revised manuscript for publication in Vaccines.

Responses to comments from reviewers

Reviewer: 2

Notes for the Authors

34    in livestock  [5]. But there are..it could be better maintain the same sentence .. in livestock, but there  

Revised accordingly (Line 36).

  1.  This suggests that when we understand how the parasite and host establish this balance, we can use appropriate methods to disrupt the balance in favor of the host to eliminate the infected T. gondii. …please explain better

Revised accordingly (Lines 42- 44).

46  infection, the host’ immune response is activated, which is critical to clear the parasites or …the host’s immune response

Revised accordingly (Line 47).

61  Taken together, TgeIF-5A plays an important role in T. gondii infection. However, whether eIF-5A affects

In summary TgelIF-5°….Nevertheless, whether…

Revised accordingly (Lines 62-63).

147combination status. We observed that rTgeIF-5A bound to the surface of Ana-1 cells, and  exhibited red fluorescence (Cy3). 148  However, on the control cells without rTgeIF-5A pro-…On the other hand ….

Revised accordingly (Line 160).

Notably, we also observed that the higher concentration of 163 rTgeIF-5A, the more obvious reduction of TLR4 levels. 164….this sentence is not understandable…..verb lost?

Revised accordingly (Lines 177-179).

318 While the molecular mechanisms of T. gondii to arrest the host cell apoptosis are not clear yet….On the other hand

Revised accordingly (Lines 350).

Reviewer 3 Report

Comments and Suggestions for Authors

The manuscript reports an in vitro investigation of the effect of the recombinant protein eIF-5A from Toxoplasma gondii on factors associated to the immune response of a murine macrophage cell line. Although the topic is relevant, some issues need to be addressed by the authors as listed below in the order of appearance in the text:

- Line 30: Write the first letter of "Toxoplasmosis" in lowercase.

- Line 46: Delete the apostrophe after "host".

- Line 69: Provide the source as well as the catalog number of Ana-1 cells.

- Line 92: Indicate the exact incubation time for the rat anti-rTgeIF-5A antibody.

- Line 98: Provide the excitation/emission wavelengths used for confocal microscopy.

- Line 139: Describe the statistical test used to assess normality of data distribution.

- Line 151: Insert scale bar in the panels of Figure 1.

- Line 153: Explicitly indicate the meaning of "Blank" and "Control" as well as the concentration of eIF-5A protein in the legend of Figure 1.

- Line 159: Note that most TLRs are not located inside the cell unlike stated (TLR1, TLR2, TLR4, TLR5, TLR6 and TLR10 are located on the cell membrane, whereas TLR3, TLR7, TLR8, and TLR9 are located in intracellular vesicles).

- Line 165: Correct the unit of eIF-5A protein concentration in Figure 2 to µg/mL as stated in Materials and Methods.

- Line 168: Explicitly indicate the meaning of "Blank" and "Control" in the legend of Figure 2.

- Line 188: Correct the unit of eIF-5A protein concentration in Figure 3 to µg/mL as stated in Materials and Methods.

- Line 192: Explicitly indicate the meaning of "Blank" and "Control" in the legend of Figure 3.

- Line 206: Correct the unit of eIF-5A protein concentration in Figure 4 to µg/mL as stated in Materials and Methods.

- Lines 208/212: Explicitly indicate the meaning of "Blank" and "Control" in the legend of Figure 4.

- Line 228: Indicate the percentage of apoptotic cells induced by rTgeIF-5A treatment.

- Line 229: Explain in Materials and Methods how early and late stages of apoptosis were distinguished in the FITC-labeled Annexin V and PI staining assay.

- Line 231: Correct the unit of eIF-5A protein concentration in the column graphs of Figure 5 to µg/mL as stated in Materials and Methods and insert the missing "e" letter of "Annxin-FITC" in the cytometric graphs.

- Line 234: Explicitly indicate the meaning of "Blank" and "Control" in the legend of Figure 5.

- Line 246: Correct the unit of eIF-5A protein concentration in Figure 6 to µg/mL as stated in Materials and Methods.

- Line 249: Explicitly indicate the meaning of "Blank" and "Control" in the legend of Figure 6.

- Line 269: Remove "which" or correct "leading" to "leads".

- Line 276: Replace "of" with "against" in "innate immunity of T. gondii".

- Line 284: Correct "was" to "were" in "there was no differences".

- Line 311: Discuss the possible reasons why 10 μg/mL eIF-5A protein was most effective in inducing the phagocytic ability of macrophages than higher concentrations such as 20, 40 and 80 μg/mL.

- Line 318: Replace "While" with "However".

- Line 338: Discuss in a more practical way how eIF-5A of T. gondii can be explored in the development of vaccines against this parasite.

Comments on the Quality of English Language

Minor editing of English language required.

Author Response

Dear reviewer:

Thank you for the constructive review and critical comments. We have uploaded the revised manuscript with highlighted changes. We believe that the manuscript is now much improved with changes made in response to the reviewers’ comments, and hopefully, we have already addressed all the comments and suggestions. Meanwhile, we have sought help from a native speaker for language editing. Thanks again for your consideration of our revised manuscript for publication in Vaccines.

Responses to comments from reviewers

Reviewer: 3

The manuscript reports an in vitro investigation of the effect of the recombinant protein eIF-5A from Toxoplasma gondii on factors associated to the immune response of a murine macrophage cell line. Although the topic is relevant, some issues need to be addressed by the authors as listed below in the order of appearance in the text:

- Line 30: Write the first letter of "Toxoplasmosis" in lowercase.

Revised accordingly (Line 31).

- Line 46: Delete the apostrophe after "host".

Revised accordingly (Line 47).

- Line 69: Provide the source as well as the catalog number of Ana-1 cells.

Revised accordingly (Line 70).

- Line 92: Indicate the exact incubation time for the rat anti-rTgeIF-5A antibody.

Revised accordingly (Line 96).

- Line 98: Provide the excitation/emission wavelengths used for confocal microscopy.

Revised accordingly (Lines 101-102).

- Line 139: Describe the statistical test used to assess normality of data distribution.

Revised accordingly (Line 148).

- Line 151: Insert scale bar in the panels of Figure 1.

Revised accordingly (Figure 1).

- Line 153: Explicitly indicate the meaning of "Blank" and "Control" as well as the concentration of eIF-5A protein in the legend of Figure 1.

Revised accordingly (Figure 1).

- Line 159: Note that most TLRs are not located inside the cell unlike stated (TLR1, TLR2, TLR4, TLR5, TLR6 and TLR10 are located on the cell membrane, whereas TLR3, TLR7, TLR8, and TLR9 are located in intracellular vesicles).

It was revised accordingly in lines 172-173s.

- Line 165: Correct the unit of eIF-5A protein concentration in Figure 2 to µg/mL as stated in Materials and Methods.

Revised accordingly (Figure 2).

- Line 168: Explicitly indicate the meaning of "Blank" and "Control" in the legend of Figure 2.

Revised accordingly (Figure 2).

- Line 188: Correct the unit of eIF-5A protein concentration in Figure 3 to µg/mL as stated in Materials and Methods.

Revised accordingly (Figure 3).

- Line 192: Explicitly indicate the meaning of "Blank" and "Control" in the legend of Figure 3.

Revised accordingly (Figure 3).

- Line 206: Correct the unit of eIF-5A protein concentration in Figure 4 to µg/mL as stated in Materials and Methods.

Revised accordingly (Figure 4).

- Lines 208/212: Explicitly indicate the meaning of "Blank" and "Control" in the legend of Figure 4.

Revised accordingly (Figure 4).

- Line 228: Indicate the percentage of apoptotic cells induced by rTgeIF-5A treatment.

Revised accordingly (Lines 246-251).

- Line 229: Explain in Materials and Methods how early and late stages of apoptosis were distinguished in the FITC-labeled Annexin V and PI staining assay.

Revised accordingly (Lines 132-134).

- Line 231: Correct the unit of eIF-5A protein concentration in the column graphs of Figure 5 to µg/mL as stated in Materials and Methods and insert the missing "e" letter of "Annxin-FITC" in the cytometric graphs.

Revised accordingly (Figure 5).

- Line 234: Explicitly indicate the meaning of "Blank" and "Control" in the legend of Figure 5.

Revised accordingly (Figure 5).

- Line 246: Correct the unit of eIF-5A protein concentration in Figure 6 to µg/mL as stated in Materials and Methods.

Revised accordingly (Figure 6).

- Line 249: Explicitly indicate the meaning of "Blank" and "Control" in the legend of Figure 6.

Revised accordingly (Figure 6).

- Line 269: Remove "which" or correct "leading" to "leads".

Revised accordingly (Line 291).

- Line 276: Replace "of" with "against" in "innate immunity of T. gondii".

Revised accordingly (Line 301).

- Line 284: Correct "was" to "were" in "there was no differences".

Revised accordingly (Line 309).

- Line 311: Discuss the possible reasons why 10 μg/mL eIF-5A protein was most effective in inducing the phagocytic ability of macrophages than higher concentrations such as 20, 40 and 80 μg/mL.

Revised accordingly (Lines 339-343).

- Line 318: Replace "While" with "However".

Revised accordingly (Line 350).

- Line 338: Discuss in a more practical way how eIF-5A of T. gondii can be explored in the development of vaccines against this parasite.

Revised accordingly(Lines 371-375).

Round 2

Reviewer 1 Report

Comments and Suggestions for Authors

The revised manuscript looks good. The authors tried their best, and answered most of my questions and concerns.